# Plant Growth-Promoting Bacterial Consortia as a Strategy to Alleviate Drought Stress in *Spinacia oleracea*

**DOI:** 10.3390/microorganisms10091798

**Published:** 2022-09-06

**Authors:** Claudia Petrillo, Ermenegilda Vitale, Patrizia Ambrosino, Carmen Arena, Rachele Isticato

**Affiliations:** 1Laboratoire de Chimie Bactérienne, Institut de Microbiologie de la Méditerranée, CNRS-Aix-Marseille University, 31 Chemin Joseph Aiguier, 13009 Marseille, France; 2Department of Biology, University of Naples Federico II, Via Cinthia, 80126 Naples, Italy; 3AGRIGES srl, 82035 San Salvatore Telesino, Italy; 4Interuniversity Center for Studies on Bioinspired Agro-Environmental Technology (BAT Center), 80055 Portici, Italy

**Keywords:** plant growth-promoting bacteria, biofertilizer, drought-stress, plant–microbe interaction, seed biopriming

## Abstract

Drought stress is one of the most severe abiotic stresses affecting soil fertility and plant health, and due to climate change, it is destined to increase even further, becoming a serious threat to crop production. An efficient, eco-friendly alternative is the use of plant growth-promoting bacteria (PGPB), which can promote plant fitness through direct and indirect approaches, protecting plants from biotic and abiotic stresses. The present study aims to identify bacterial consortia to promote *Spinacia oleracea* L. cv Matador’s seed germination and protect its seedlings from drought stress. Eight PGPB strains belonging to *the Bacillus, Azotobacter*, and *Pseudomonas* genera, previously characterized in physiological conditions, were analyzed under water-shortage conditions, and a germination bioassay was carried out by biopriming *S. oleracea* seeds with either individual strains or consortia. The consortia of *B. amyloliquefaciens* RHF6, *B. amyloliquefaciens* LMG9814, and *B. sp*. AGS84 displayed the capacity to positively affect seed germination and seedlings’ radical development in both standard and drought conditions, ameliorating the plants’ growth rate compared to the untreated ones. These results sustain using PGPB consortia as a valid ameliorating water stress strategy in the agro-industrial field.

## 1. Introduction

Plants generally experience many abiotic stresses during their growth, such as heat, drought, salinity, and acidity [1]. Among these injuries, drought is considered one of the most severe environmental stresses affecting soil fertility, plant health, and therefore crop yield. It occurs due to temperature dynamics, light intensity, and low rainfall [2] and impacts all of the main agricultural lands [3]. It has been estimated that in the last four decades, drought caused a reduction in cereal yield by up to 10% and will affect crop production in over 50% of cultivated land by 2050 [4]. As water plays a crucial role in a plant’s vital processes [4], drought stress negatively affects seed germination efficiency and seedling growth and decreases plants’ biomass by reducing leaves’ size and number and roots’ development [5,6]. Plants are able to face numerous abiotic stress factors through several strategies, which imply different morphological and physiological strategies [7,8,9,10]. Nevertheless, anthropogenic activities, together with global warming, lead to the increased severity of droughts, imposing a severe threat on agricultural productivity [2]. The inocula of drought-tolerant plant growth-promoting bacteria (PGPB) has been proposed as an eco-friendly strategy to limit drought damage and reduce chemical fertilizers’ overuse [11,12,13,14,15,16]. PGPB are microorganisms naturally capable of enhancing plants’ growth by direct and indirect approaches, comprising the production of phytostimulant metabolites, such as hormones or siderophore; the promotion of plants’ nutrients uptake; or the inhibition of pests [17,18]. More importantly, they can arrange beneficial associations with the roots of plants to improve their growth and increase tolerance to abiotic stresses, such as water shortage [13]. In particular, drought-tolerant PGPB (DT-PGPB) have been shown to increase plant growth in water-limited conditions compared to sensitive stress PGPB strains. Under drought stress, DT-PGPB produce phytohormones, such as indole-3-acetic acid (IAA), abscisic acid (ABA), gibberellins (GA), and cytokinins (CKs), that can reduce stress damage and improve plant growth. For example, it has been reported that IAA produced by some PGPB can alleviate plant drought stress by inducing lateral root development and therefore water uptake [19]. More recently, it has been demonstrated that some DT bacteria, such as *Pseudomonas azotoformans* FAP5, are capable of producing exopolysaccharides (EPS), forming biofilm, and P solubilization under water stress conditions [20]. Wheat plants inoculated with *P. azotoformans* FAP5 showed increased plant growth, photosynthetic pigment, physiological attributes, and antioxidative enzymatic activities under drought stress [21]. Recently, importance has been given to applying PGPB consortia, groups of bacteria exhibiting complementary features [1]. Indeed, bacterial consortia were shown to have higher performance than the inoculation of individual species to promote plant drought tolerance [22]. For example, a consortium of *Bacillus subtilis, Bacillus thuringiensis, and Bacillus megaterium* has been shown to significantly enhance metabolic activities, including the chlorophyll content, riboflavin, L-asparagine, and aspartate accumulation, and decrease the stress response activities in *Cicer arietinum* under drought stress [23]. Likewise, a consortium including *Ochrobactrum pseudogrignonense* (RJ12), *Pseudomonas sp.* (RJ15), and *B. subtilis* (RJ46) was capable of N fixation, P solubilization, the production of IAA, and siderophores, offering drought stress tolerance to *Vigna mungo* and *Pisum sativum*, through promoting seed germination, root and shoot length, and chlorophyll contents [23].

The purpose of this work was to identify DT-PGPB among a collection of eight bacteria belonging to the genera *Bacillus*, *Azotobacter*, and *Pseudomonas* [17,18]. To this aim, all the strains were characterized for PGP traits and the ability to promote *Spinacia oleracea* L. cv Matador germination under simulated drought stress. Although spinach germination occurs routinely in seedbeds without water stress, this plant was selected as a model plant since it is very sensitive to water stress [24] and allowed us to examine the DT-PGPB influence at its different developmental stages in vitro (germination and seedling growth).

A consortium of three DT bacterial strains was identified as *B. amyloliquefaciens* RHF6, *B. amyloliquefaciens* LMG9814, and *B. sp.* AGS84 and emerged as the most promising, positively affecting *S. oleracea*’s seed germination efficiency and promoting the seedlings’ radical development.

## 2. Materials and Methods

### 2.1. Bacterial Strains and Growth Conditions

The PGPB used in this study are listed in Table 1, grown on TY medium for routine use and pure cultures stored at −80 °C into glycerol stocks [25]. Some of the strains were deposited in the culture collection of Agriges s.r.l. (San Salvatore Telesino, Benevento, Italy).

### 2.2. Phenotypic Characterization and Growth Conditions

The phenotype of the bacterial strains was determined by visual inspection. The facultative anaerobic growth was determined using the AnaeroGen sachets (Unipath Inc., Nepean, ON, Canada) placed in a sealed jar with bacteria streaked on TY agar plates and incubated at 37 °C for 3 days. To confirm the sporulation ability, the bacterial strains were grown in Difco sporulation medium (DSM) (8 g/L Nutrient broth No. 4, 1 g/L KCl, 1 mM MgSO_4_, 1 mM Ca(NO_3_)_2_, 10 μM MnCl_2_, 1 μM FeSO_4_ (Sigma-Aldrich, Taufkirchen, Germany). The optimum growth conditions were determined by growing the strains in TY agar at different pH (2.0, 4.0, 6.0, 7.0, 8.0, 10.0, 12.0) [26], temperatures (4, 15, 25, 37, 50, 60 °C) [27], and PEG 6000 (0, 5, 10, 15, 20%) ranges.

### 2.3. Bioassays for PGP Traits

The eight strains were characterized for their PGP traits as described below. When drought stress is simulated, 15% PEG 6000 is supplemented to the media [28].

#### 2.3.1. Biofilm Production and Swarming Motility

To investigate the capacity of producing biofilm, bacterial isolates were grown in 24-well culture plates in TY broth for 48 h, in static conditions at 37° [29]. After that, the supernatant was discarded, adhered cells were rinsed three times with distilled water, and 1 mL of a 0.1% Crystal Violet (CV) solution was added to stain the adhered biomass. Plates were incubated for 30 min at room temperature, carefully washed three times with distilled water, and patted dry. Dye attached to the wells was extracted with 1 mL of 70% ethanol and quantified at an absorbance of 570 nm. Data were normalized by total growth estimated by OD_600 nm_. The experiment was performed in triplicate. Swarming motility was assayed according to the method described by Petrillo et al. [18]. TY agar 0.7% plates were spot inoculated with 3 μL of the freshly grown bacterial culture (10^7^ CFU/mL). After overnight incubation at 37 °C, the swarm diameters were measured.

#### 2.3.2. Phosphate Solubilization

The microbial ability to solubilize phosphate was evaluated by the spot inoculation of 3 μL of a freshly grown bacterial culture (10^7^ CFU/mL) onto Pikovaskya’s agar medium as described by Petrillo et al. [18]. The plates were incubated at 28 °C for 10–15 days. A positive result is represented by the formation of transparent zones around the bacterial colonies [30].

#### 2.3.3. Indole-Acetic Acid (IAA) Detection

The IAA production was measured as described by Damodaran [31], with some modifications. Briefly, each strain was cultured in 10 mL of TY broth at 37 °C for 4 days with shaking at 150 rpm. Then, 1 mL of bacterial supernatant was mixed with 2 mL of Salkowski reagent (0.5 M FeCl_3_ in 35% HClO_4_ solution), and the solution was vortexed and incubated at room temperature for 30 min. The formation of a pink color represented a positive reaction [31]. The quantitative estimation of IAA (μg/mL) was achieved by recording spectroscopic absorbance (Horiba Scientific, Edison, NJ, USA) at 535 nm using a standard curve prepared with pure IAA (Sigma) in the range 0–100 μg/mL [32]. Sterile TY broth was used as a control.

#### 2.3.4. Ammonia Production

To detect ammonia’s production, the eight bacteria were grown in 4% peptone broth and incubated for seven days at 30 °C. After that, 0.5 mL of Nessler’s reagent was added to the bacterial suspension. The development of a brown to yellow color indicates ammonia production. The samples’ absorbance was measured at 450 nm using a spectrophotometer (Nova Spec II, Pharmacia, Burladingen, Germany). The quantitative estimation of the amount of ammonia production by the bacterial strains was performed by comparing the results with a standard curve generated using a standard ammonium sulfate solution.

#### 2.3.5. Siderophores Production

The siderophores’ production was determined through the Chrome Azurol S (CAS) assay [33]. An amount of 3 mL of freshly grown bacterial cultures was spot inoculated on CAS agar plates and incubated at 28 °C. The appearance of a yellow-orange halo zone around the bacterial colonies was a positive indicator of siderophores production and the halos’ diameters were measured after 4 days of incubation.

#### 2.3.6. Biosurfactants Production

The bacterial isolates were spot inoculated on blood agar plates (BBL™ Trypticase™ Soy Agar (TSA II) supplemented with 5% Horse Blood), and after 72 h of incubation at 28 °C, the clear zone around the colonies indicates a positive result [34].

#### 2.3.7. Screening for Hydrolytic Enzymatic Activity

The eight bacterial strains were grown separately in 5 mL of TY broth at 37 °C overnight with shaking at 150 rpm. An amount of 3 µL of each fresh bacterial culture was spot inoculated on plates containing different carbon sources to test hydrolytic enzyme activity. The protease activity was assayed on Skimmed Milk Agar (SMA) [35]. After overnight incubation at 37 °C, the formation of a clear halo around the colonies was considered a positive activity. To detect the amylase activity, the method described by Alvariva with Starch Agar plates was used [36]. After the overnight incubation at 37 °C, the plates were flooded with iodine solution, and the hydrolysis of starch was observed as a colorless zone around the colonies. To detect cellulase and xylanase activities, Xylanase Production Medium (XPM) agar plates with 0.5% xylan (Megazyme) [37] and a minimal medium with 0.5% carboxymethylcellulose (CMC) [38] as sole carbon sources, were used. The plates were incubated at 37 °C for 3 days after which hydrolysis zones were visualized by flooding the plates with 0.1% Congo Red for 15–20 min and then 1 M NaCl was used to destain them by washing twice. Plates without CMC and xylan were used as no substrate controls. Transparent hydrolytic zones around the colonies were considered positive.

To quantify the activity observed on a plate, the ratio of the clear zone diameter to colony diameter was measured, assuming the largest ratio represents the highest activity. Hence, the following formula was applied:$$\%\;Efficiency=\frac{total\;diameter-colony\;diameter\;}{colony\;diameter}\times 100$$

All experiments were performed in triplicate.

#### 2.3.8. Catalase Assay

The quantitative determination of catalase activity of bacterial strains was evaluated by following the loss of absorbance at 240 nm as previously described [39]. Briefly, 1 mL of bacterial culture grown for 48 h at 37 °C in TY broth- 15% PEG 6000 was incubated for 30 min at room temperature in 1 mL of hydrogen peroxide solution (50 mM Potassium Phosphate Buffer, pH 7.0, 0.036% (*w*/*w*) H_2_O_2_). After that, the samples were centrifuged for 1 min at 13,000× *g*, and the hydrogen peroxide concentration in the solution was determined by measuring the absorbance at 240 nm. The amount of peroxide removed was calculated as reported below:$$\%H_{2}O_{2}\;removed=\frac{1-Asamples\;}{Acontrol}\times 100$$
where *Acontrol* is the absorbance of 1 mL of hydrogen peroxide solution.

#### 2.3.9. DPPH Assay

The α,α-diphenyl-β-picrylhydrazyl (DPPH) free radical scavenging method was used to evaluate the potential antioxidant activity of the bacterial strains as described by Mazzoli et al. [40], with some modifications. The bacterial strains were grown in TY broth supplied with 15% PEG 6000 and incubated for 48 h at 37 °C with shaking at 150 rpm. An amount of 1mL of the culture was vigorously mixed with 1 mL of 0.2 mM DPPH solution dissolved in ethanol and incubated at 25 °C for 30 min in the dark. The DPPH radical scavenging activity was evaluated by measuring the absorbances of the supernatants at 517 nm, according to the following equation:$$DPPH\;radical\;scavenging\;activity\;\left(\%\right)=\frac{1-Asamples\;}{Acontrol}\times 100$$
where *Asample* is the absorbance of the reacted mixture of DPPH with the sample, and *Acontrol* is the absorbance of the DPPH solution.

### 2.4. Germination Assay

To test the ability of the microbial strains to promote seeds’ germination, the bacterial strains were cultured overnight in TY medium at 37 ± 2 °C (25 ± 2 °C for strain AGS54). Then, the cells’ concentration (CFU/mL) was determined by a Burker chamber and diluted to 1 × 10^8^ CFU/mL in 1X Phosphate-Buffered Saline (PBS). For the consortia, the dilutions of the single strains were mixed, keeping a 1:1:1 ratio. Seeds of *S. oleracea* L. cv Matador were sterilized with 5% H_2_O_2_ for 15 min and rinsed with sterile deionized water [41]. Then, 45 seeds were incubated with a dilution of each one of the strains adjusted to 1 × 10^8^ CFU/mL with 1X PBS (single or consortium) for about 4 h at room temperature under stirred conditions to promote the bacterial adhesion to the seeds. Seeds treated with 1X PBS were used as control. The treated seeds were then spread on a 1.8% water agar medium (WA) and incubated at 20 °C in dark conditions for about one week to let germinate. Germination was defined as the protrusion of radicles through the seed coat and the number of germinated seeds was counted daily. The germination rate and efficiency were obtained from three independent experiments, for a total of 135 observations/treatments (15 seeds × 3 Petri dishes × 3 replicates). To determine the seedlings’ well-being, the length of primary roots was also evaluated by ImageJ software, using pictures of the plates and a ruler taken at the same height. Then, the length in pixels was converted into centimeters by using the ruler as a scale. Spinach radicles’ well-being was evaluated by observing them with a stereoscopic microscope with 10× magnification [41].

### 2.5. Adhesion Assay

To evaluate bacterial adhesion onto *S. oleracea*’s seeds (each of the different treatments and the control), a modified method described by Hashmi was performed [42]. Three seeds were randomly collected to count bacterial cells adhering to their surface by flow cytometry. The seeds of each treatment were placed in sterile tubes containing 1 mL of sterile 1X PBS and vortexed vigorously for 1 min.

### 2.6. Microbial Compatibility In Vitro 

To assess the ability of the eight strains to coexist, they were subjected to an in vitro compatibility test using the agar diffusion assay [43]. In particular, a single colony of each strain was inoculated in TY medium and incubated at 37 ± 2 °C for ~18 h, 150 rpm. An amount of 100 μL of each strain was plated on a TY agar medium, and 5 μL of the other strains were spotted on top of it. The plates were then incubated at 37 ± 2 °C. The microorganisms that overlap are considered compatible. On the other hand, when an inhibition halo appears, the two microorganisms are considered incompatible.

### 2.7. Drought Stress Treatment

To evaluate the bacterial consortium efficiency of protecting *S. olearacea*’s growth from drought stress, the method described by Muchate was adopted, with some modifications [44]. Actively growing shoot cultures were transferred to the Murashige and Skoog (MS) basal nutrient medium supplemented with 5% PEG 6000 prior to root-dipping in the bacterial suspension, where single strains were mixed, keeping a 1:1:1 ratio. The negative controls’ roots were dipped in 1X PBS. MS medium pH was adjusted to 5.8 and supplied with 0.7% agar prior to autoclaving at 121 °C for 15 min. All the cultures were incubated in sterile jars and grown at 25 ± 2 °C temperature, a 16h photoperiod using cool white fluorescent light (40 μM m^−2^ S^−1^ irradiance), and 70% relative humidity for 27 days. After that, the total and partial (ipogeal/epigeal) length and weight were measured.

### 2.8. Fluorescence Emission Measurements and Photosynthetic Pigment Determination

Chlorophyll *a* Fluorescence emission measurements were carried out on three leaves per treatment using a portable fluorometer (FluorPen FP100max, Photon System Instruments, Brno, Czech Republic), equipped with a light sensor (Photon System Instruments, Brno, Czech Republic). The basal fluorescence signal, Fo, was induced by an internal LED blue light (1–2 μmol photons m^−2^ s^−1^) on 30′ dark-adapted leaves. The maximal fluorescence signal in the dark, Fm, was determined by a saturating light pulse of 3000 μmol photons m^−2^ s^−1^ of 1s. The PSII maximal photochemical efficiency, Fv/Fm, was calculated as the ratio (Fm − Fo)/Fm. After fluorescence measurements, the same leaves (three per treatment) were collected to determine total chlorophylls (a + b), total carotenoids (x + c), a/b and (a + b)/(x + c) ratios. Fresh samples (10 mg) were extracted using ice-cold 100% acetone and centrifuged (Labofuge GL, Heraeus Sepatech, Hanau, Germany) at 5000 rpm for 5 min. The absorbance of supernatants was measured by spectrophotometer (Cary 100 UV-VIS, Agilent Technologies, Santa Clara, CA, USA) at 470, 645, and 662 nm. The pigment concentration was determined using Lichtenthaler equations [45] and expressed in mg g^−1^ of fresh weight (mg g^−1^ FW). All the measurements were performed after 27 days of growth. 

### 2.9. Statistical Analysis

All the statistical analyses were performed using GraphPad Prism 8 software. Data were expressed as mean ± SD. Differences among groups were compared by ANOVA, Tukey’s, or Dunnett’s tests as indicated in figure legends. Differences were considered statistically significant at *p* < 0.05. Plants’ physiological data were analyzed by applying the one-way analysis of variance (ANOVA) and Duncan’s multiple comparison test post hoc with a significance level of *p* < 0.05. The Shapiro–Wilk test was applied to check the normal distribution of data. Values are reported as mean values ± standard error (*n* = 3).

## 3. Results 

### 3.1. In Vitro Characterization of Potential DT-PGPB

The six bacterial strains provided by Agriges S.r.l., listed in Table 1, were preliminarily characterized for growth properties (Appendix A) and compared with two *Bacilli*, strains RHF6 and RHFS10, which recently emerged as promising PGPB [17,18]. All the strains used in this study represent well-recognized PGPB genera, with more than 70% identified as members of the *Bacillus* genus. In contrast, strains LS132 and AGS54 were identified as the Gram-negative *A. chroococcum* and *P. fluorescens* (Table 1).

The eight strains can be classified as facultative anaerobic; almost all of them fit in the mesophile group, except for strains LMG9814, AGS84, and AGS108, which can grow up to 60 °C, and strain AGS54, which grows between 4 and 40 °C (Appendix A). To determine the tolerance to water deficiency, the eight strains were grown in the presence of different PEG 6000 concentrations (Section 2). A total of 60% of the strains tolerate up to 15% PEG 6000, while AGS172, AGS84, and AGS54 strains survived up to 20%. To compare the PGP potential of the six new strains to the already-characterized Bacillus strains RHF6 and RHFS10, their ability to produce growth hormones and siderophores, solubilize phosphorus, and the capability of hydrolyzing different polymers was assayed (Appendix A). Most of the strains are potentially able to colonize root apparatus since they are capable of surface spreading by swarming and forming biofilm (Appendix A) [46]. At the same time, only five were found positive for biosurfactant production. Strain AGS54 is the best IAA producer, followed by AGS84 and AGS172. On the other hand, strain LS132 releases the highest amount of ammonia, as expected of an *Azotobacter* [47]. 

Based on this preliminary characterization, it is possible to say that this bacterial collection has a strong PGPB potential in vitro.

### 3.2. Characterization of PGP Traits under Drought Stress Conditions

To identify potential DT-PGB, further characterization of the bacterial strains was repeated under drought stress conditions in the presence of 15% PEG 6000. As expected, the PGP performances of all the strains were lower than those registered in physiological conditions (Table 2, Appendix A).

The most impressive loss was observed for the hydrolytic activities and cellulolytic activity on top of all. Strain RHF6, which exhibited one of the highest hydrolytic potentials, lost it completely, together with strains LS132 and AGS54. On the contrary, the xylanase activity exhibited by strains AGS172 and AGS108 and the amylase activity exhibited by strains AGS84 and AGS108 increased under drought-stress conditions [48]. The same tendency was observed for the IAA production shown by strains RHFS10 and LS132, which increased almost three and two times, respectively, reaching 18 and 2.4 µg/mL (Table 2). 

### 3.3. Antioxidant and Scavenging Activity

PGPB can enhance plants’ oxygen radical scavenging by producing antioxidant enzymes [49]. For this purpose, the bacterial antioxidant activity was also evaluated. In particular, strains’ capacity to degrade hydrogen peroxide into water and oxygen and the α,α-diphenyl-β-picrylhydrazyl (DPPH) radical scavenging activity were tested (Appendix A).

The catalase activity increased under PEG 6000-simulated drought stress in a dependent manner in almost 90% of the tested strains, with strain LS132 showing the best response. In contrast, the opposite tendency was observed for strain AGS108 only, displaying a slight reduction in the enzymatic activity under hydric stress (Appendix A). Instead, the free radical scavenging activity does not seem to be affected by the lack of moisture, and only strain RHFS10 was most influenced by this condition (Appendix A).

### 3.4. Effects of Seed-Biopriming on S. oleracea Germination In Vitro 

To confirm the DT-PGPB potentiality of the selected strains, a germination bioassay was carried out on the early vegetative growth stage of *S. oleracea* seedlings, as reported in the Section 2. As reported in Figure 1, the selected bacteria affected seed germination with different efficiencies.

Seed-biopriming using strains RHF6, LMG9814, and AGS84 significantly improved seeds’ germination rate and efficiency and produced the healthiest seedlings against the untreated control seeds (Figure 1A). Strain AGS108 also positively affected the seeds’ germination. In particular, the most extended radicle length (6.82 cm) and the highest germination rate and efficiency (Figure B–D) were recorded for seeds bioprimed with *B. amyloliquefaciens* strain RHF6. On the other hand, strains RHFS10-, LS132-, AGS172-, and AGS54-bioprimed seeds exhibited lower viability and vigor (Figure 1A). A possible explanation for the outstanding effects exerted by strains RHF6, LMG9814, and AGS84 could be linked to the stronger adhesion of bacterial cells to seeds (Figure 2).

To evaluate this parameter, three bioprimed seeds were randomly collected to count bacterial cells adhering to their surface by flow cytometry, as described in the Section 2. Once again, strain RHF6 exhibited the best performance. 

### 3.5. Effects of Bacterial Consortia on S. oleracea Germination In Vitro 

A more recent strategy to increase plant growth is the application of consortia of PGPB exhibiting complementary traits [1]. Indeed, bacterial consortia were shown to have higher performance than the inoculation of individual species [50]. To construct the bacterial consortia, an in vitro compatibility test was performed by inoculating one strain at a time on a lawn culture of each other strain from the collection, as described in the Section 2. Based on the results reported in Appendix A, four consortia, named C1, C2, C3, and C4 were prepared out of the eight potential PGPB (Table 1). A germination assay was performed to verify the action of the consortia on the germination phase of *S. oleracea* by imbibing the seeds with the four culture mixes (adjusted to 1 × 10^8^ CFU/mL, maintaining a 1:1:1 ratio of the single strains) (Figure 3A).

As previously described, the seedlings’ well-being was evaluated through several parameters (Figure 3B–D). Out of the four consortia, C2 made of strains RHF6, AGS84, and LMG9814, gave the best results, increasing the germination rate and efficiency up to ~100% and producing seedlings with the longest primary roots (6.96 cm) (Figure 3C). The PGPB treatment also affected the number of lateral radicles observed with a stereoscopic microscope with 10× magnification (Appendix A). Interestingly, the best consortium brings together the PGPB, which showed the strongest effect when assayed individually (Figure 1). For this reason, consortium C2 was chosen for further analysis.

### 3.6. Drought Stress Treatment

To assess the ability of the best consortium to protect *S. oleracea* shoots from drought stress, germinated seedlings from the previous assay, pre-treated with 1X PBS (Control) or C2, were moved to jars containing a thin layer of Murashige and Skoog (MS) medium, supplied or not with 5% PEG6000 to simulate drought stress (Figure 4A). Before growing the seedlings for 27 days in sterile jars, the seedlings were inoculated with C2 by dipping their roots into the bacterial suspension or in 1X PBS as control (Section 2*)*.

After incubation, the shoots were examined by measuring the length and weight of the ipogeal and epigeal biomass and the total seedling biomass (Figure 4B,C). As expected, the water stress had a clear negative impact on the general well-being of *S. oleracea*’s control plants (untreated), which was interestingly reversed by the PGPB treatment. As reported in Figure 4B, the epigeal length increased by 40% by bacterial inoculation in both conditions. The beneficial effect of the C2 was more evident for the ipogeal system under stress conditions, with an increase of 65% of the length. The treatment of spinach plants with C2 also determined an increase in epigeal biomass (about 20%), again more evident in drought conditions (Figure 4C). No effect was exerted by bacteria inocula on ipogea biomass in both conditions.

The water stress negatively affected the photosynthetic pigment content of spinach plants (C-_PEG6000_), inducing lower concentrations (*p* < 0.05) of total chlorophylls and carotenoids compared to untreated plants (C-) (Figure 5A,C). The PGPB treatment reversed this tendency, increasing the pigments (*p* < 0.05) in C2 plants even under drought stress (C2 _PEG6000_) (Figure 5A,C). Untreated and stressed control plants showed a reduced Chl a/b ratio which reached the lowest (*p* < 0.05) value under drought stress. On the contrary, the PGPB promoted (*p* < 0.05) the ratio a/b in the absence and presence of drought (Figure 5B). C2 and C-_PEG6000_ exhibited a reduced (*p* < 0.05) Chl/Car ratio compared to untreated plants (C-), but the PGPB treatments, in the presence of drought (C2 _PEG6000_), determined a value comparable to the control (Figure 5D). Finally, all treatments showed a higher (*p* < 0.05) Fv/Fm ratio than the untreated control (Figure 5E).

## 4. Discussion

The application of PGPB to the agricultural field is considered to have the potential for improving plant growth in extreme environments characterized by water shortages [51,52]. The ability of these beneficial bacteria to survive and compete with the soil microflora and colonize the rhizosphere remains a critical step for their successful application, especially in dry soil [53]. For these reasons, the application of DT-PGPB may represent a valid strategy to deliver beneficial effects on plants. The present study proposes the analyses of eight bacteria from the *Bacillus*, *Pseudomonas*, and *Azotobacter* genera to identify potential DT-PGPB able to alleviate water-shortage-induced stress on *S. oleracea* seedlings and plants. All the analyzed strains can grow up to 15% PEG 6000 (about −0.36 MPa water potential) to a temperature of 40 °C and at a wide pH range, suggesting natural adaptation to drought. These features are necessary to withstand and proliferate in dry soils since drought conditions are frequently accompanied by an increase in temperature and changes in soil pH. The strains were also analyzed for beneficial traits, such as P solubilization and IAA and ammonia production, in a simulated water shortage. While some activities, such as cellulolytic activity, were lower than the ones registered in optimal conditions (Table 2), in some strains, the xylanase (strains AGS172 and AGS108) and amylase (AGS84 and AGS108) activities increased under drought-stress conditions. This behavior agrees with what has been recently stated by Bouskill et al. [48]. Indeed, it was observed that bacterial communities could respond to water stress by increasing the hydrolytic activity of classes of enzymes correlated to the metabolism of complex C-sources. As expected, drought stress also triggered IAA production and antioxidant activity, which increased almost thrice and twice compared to the standard condition. In particular, phytohormone production reached 18 and 2.4 µg/mL in strains RHFS10 and LS132, respectively (Table 2). Hence, a germination bioassay was performed to evaluate the effects of the potential DT-PGPB on the early vegetative growth stage of *S. oleracea* seedlings. As shown in Figure 1, bacterial inoculation can exert different effects on seed germination. In some cases, it leads to an acceleration of the radicle emergence with positive outcomes on root development after the initial, rapid protrusion from the seed. In particular, *B. amyloliquefaciens* RHF6 and *B.* sp. AGS84 were able to increase the germination efficiency by about 30% and 15%, respectively, concerning the untreated seeds. This beneficial effect, especially the one shown by strain RHF6, agrees with the stronger adhesion of the bacterial cells to the seed surface (Figure 2). Other strains had a negative impact on seed germination, with a reduction in germination efficiency of about 30% for RHFS10 and AGS54 strains and 42% for AGS172 despite the high IAA amount produced by these strains (Table 2). Again, a possible explanation could be the low adhesion to the seeds. According to the in vitro compatibility, four consortia were prepared out of the eight bacterial strains. Again, the germination efficiency, rate, and primary root length were considered to determine the seedlings’ vigor (Figure 3). Interestingly, the best consortium (C2) is made of the three PGPB that exhibited the strongest beneficial effect on germination individually (Figure 1): strains RHF6, LMG9814, and AGS84. This outcome confirms the former results and allows us to hypothesize a beneficial synergic action of the three strains in the consortium C2. The same consortium was used to pre-treat *S. oleracea* seedlings subjected to drought stress. By comparison of macroscopic and physiological parameters, it was possible to highlight the bacterial capacity to protect spinach from water shortage in vitro. Impressively, the plants grown under drought stress and pre-treated with DT-PGPB seemed even healthier than the ones developed without PEG6000. This phenomenon could be due to the induced bacterial IAA production and antioxidant activities. As previously reported, bacteria produce IAA under drought conditions to cope with osmotic stress. Simultaneously, IAA produced by PGPB induces an increase in the root surface area through an increasing root tip number. These modifications in root system architecture, observable in Appendix A, could increase plant water and nutrient acquisition, reducing the water deficit [51,52]. As a result, improving the water flow from the root to shoot system might have avoided the stomatal closure. The consequent favored uptake of CO_2_, P solubilization, and IAA and siderophores production by bacteria stimulated the plant growth in terms of root length and biomass accumulation (Figure 4, Appendix A). Compared to untreated control plants, water deprivation significantly reduced the photosynthetic pigments, likely interfering with the pyrrole biosynthesis pathway necessary for chlorophylls’ production [54]. However, lower Chl a/b and a + b/x + c ratios and higher F_v_/F_m_ suggest that C2 _PEG6000_ plants modulated the production of the pigments within the antenna complexes to optimize the light-harvesting and improve the efficiency of the photosystem II (Figure 5). By improving nutrient uptake (recognized as the pivotal role of PGPB), the consortium enhanced the whole physiological status of spinach plants even under water shortage. The consortium promoted the synthesis of chlorophylls and carotenoids, which exert the critical role of non-enzymatic antioxidant compounds, especially under stressful conditions. Finally, the higher F_v_/F_m_ observed in inoculated plants compared to the control confirmed that the consortium may confer spinach plants a high tolerance to drought. 

The results obtained in this study were promising, thus sustaining the use of DT-PGPB consortia as an eco-friendly approach to increase plant performance under drought. Nevertheless, further analysis is required to improve this strategy for commercial use in the agro-industrial field of arid and semi-arid regions. 

## Figures and Tables

**Figure 1 microorganisms-10-01798-f001:**
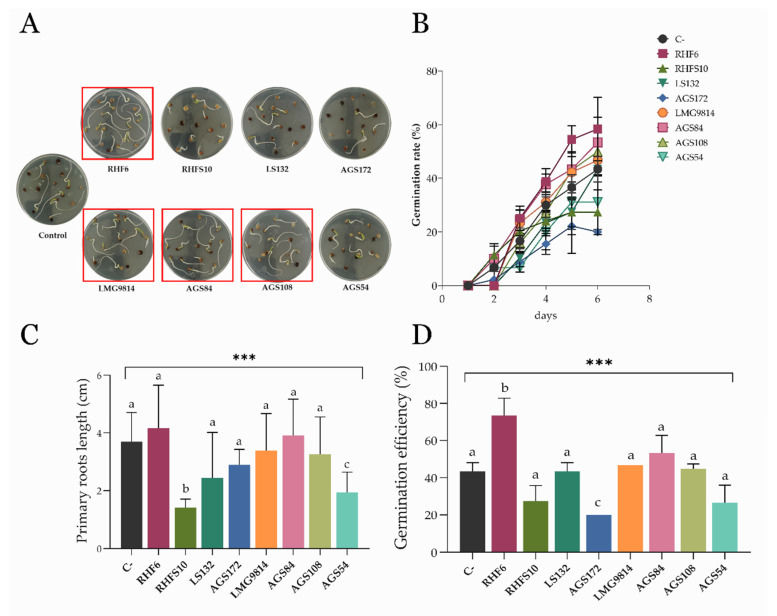
(**A**) Effects of seed-biopriming on *S. oleracea*. The red squares point out the treatments that gave the best results; (**B**) seed germination rate (%) measured over 6 days; (**C**) measure of the seedlings’ primary roots length by ImageJ software; (**D**) seed germination efficiency measured over 6 days—the comparison between the number of total germinated seeds over the number of total seeds on each plate is reported in percentage. Data are presented as means ± standard deviation (n = 3). For the comparative analysis of groups of data, one-way ANOVA was used, and *p*-values are shown in the figure: ***: extremely significant < 0.001. Different letters in each average indicate differences according to Dunnett’s test (*p* < 0.05).

**Figure 2 microorganisms-10-01798-f002:**
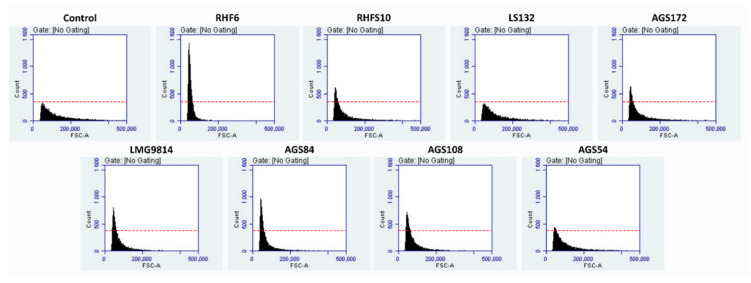
Adhesion assay. Flow cytometry analysis of *S. oleracea*-bioprimed seeds. Seeds treated with individual bacterial strains were collected randomly to count the bacterial cells adhering to their surface. In each panel, the number of cells counted (*Y*-axis) against their dimension (*X*-axis) is indicated. As a control experiment, seeds treated with 1X PBS were analyzed, and the results are reported as red dashes.

**Figure 3 microorganisms-10-01798-f003:**
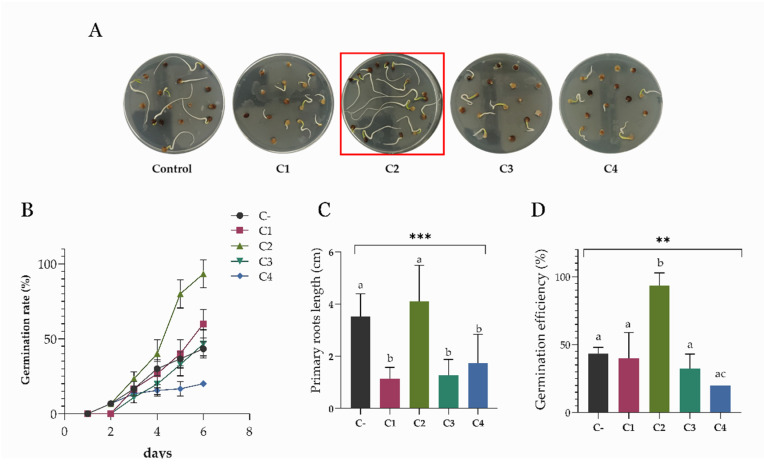
(**A**) Effects of seed biopriming with the bacterial consortia on *S. oleracea*. The red square points out the consortium that gave the best results: C2; (**B**) Seed germination rate (%) measured over a 6-day period; (**C**) measure of the seedlings’ primary roots length by ImageJ software; (**D**) seed germination efficiency measured over 6 days—the comparison between the number of total germinated seeds over the number of total seeds on each plate is reported in percentage. C1: RHFS10, AGS172, AGS108; C2: RHF6, AGS84, LMG9814; C3: RHFS10, RHF6, AGS172; C4: RHFS10, AGS54, LS132. Data are presented as means ± standard deviation (n = 3). For the comparative analysis of data groups, one-way ANOVA was used, and *p*-values are presented in the figure: ***: extremely significant < 0.001; **: significant < 0.006. Different letters in each average indicate differences according to Tukey’s test (*p* < 0.05).

**Figure 4 microorganisms-10-01798-f004:**
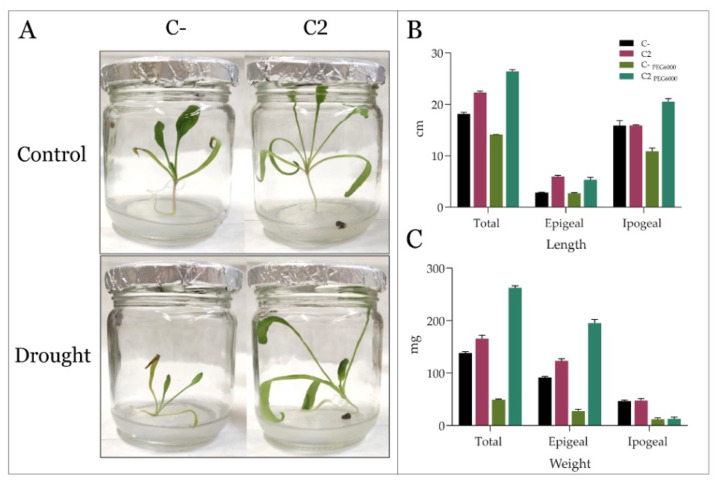
Effect of drought stress on the growth of in vitro cultures of *S. oleracea*. (**A**) The in vitro shoots, treated with 1X PBS (C-) or the bacterial consortium (C2), were cultured onto MS medium (0.7% agar) supplemented with +/− 5% PEG6000. The shoots’ total, epigeal and ipogeal length (**B**), and weight (**C**) were evaluated after 27 days of growth.

**Figure 5 microorganisms-10-01798-f005:**
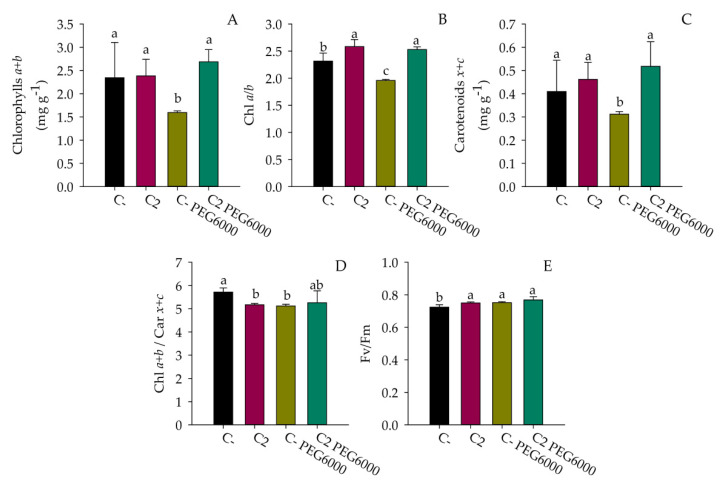
Effect of drought stress on the PSII maximal photochemical efficiency of *S. oleracea*. Total chlorophylls (a + b) (**A**), chlorophyll a/b ratio (**B**), total carotenoids (x + c) (**C**), chlorophylls/carotenoids ratio (a + b/x + c) (**D**), and PSII maximal photochemical efficiency (F_v_/F_m_) (**E**) of spinach leaves. The in vitro shoots, treated with 1X PBS (C-) or the bacterial consortium (C2) represent the unstressed plants, and those cultured onto MS medium (0.7% agar) supplemented with +/− 5% PEG6000 represent the respective stressed plants (C-_PEG6000,_ C2 _PEG6000)._ The measurements were performed after 27 days of growth. Data are presented as means ± standard deviation (n = 3). For the comparative analysis of data, one-way ANOVA was used. Different letters in each average indicate differences according to Duncan’s test (*p* < 0.05).

**Table 1 microorganisms-10-01798-t001:** List of the bacterial strains used in this study.

Strain	Species	Source	Citation
**RHF6**	*B. amyloliquefaciens*	Sand (Spain)	[18]
**RHFS10**	*B. vallismortis*	Rhizosphere (Spain)	[17]
**LS132**	*A. chroococcum*	Rhizosphere (Italy)	Agriges collection
**AGS172**	*B. subtilis*	*Unknown*	Agriges collection
**LMG9814**	*B. amyloliquefaciens*	Soil	Agriges collection
**AGS84**	*Bacillus.* sp.	Grape leaves	Agriges collection
**AGS108**	*B. amyloliquefaciens*	*Unknown*	Agriges collection
**AGS54**	*P. fluorescens*	Soil	Agriges collection

**Table 2 microorganisms-10-01798-t002:** Summary of plant growth-promoting traits exhibited by the eight bacterial strains under drought stress.

	PGPR Traits					Hydrolytic Activities (%)			
Strain	PVK	IAA (µg/mL)	Ammonia Production (mg/L)	Siderophores (%)	Biosurfactants	Protease	Amylase	Xylanase	CMC
**RHF6**	+	4.2 ± 0.10	1.3 ± 0.02	3.5	++	0	0	0	0
**RHFS10**	+	18 ± 0.03	0.0	25	+++	0	50	75	0
**LS132**	−	2.4 ± 0.03	1.8 ± 0.01	0	+	0	0	0	0
**AGS172**	+	5.2 ± 0.02	1.7 ± 0.01	5.2	++	100	50	75	0
**LMG9814**	−	3.5 ± 0.09	0.8 ± 0.09	3.2	++	100	66.7	0	0
**AGS84**	+	4.4 ± 0.03	1.1 ± 0.02	4.6	++	0	100	75	0
**AGS108**	+	2.3 ± 0.05	0.6 ± 0.01	3.5	+	100	75	75	0
**AGS54**	−	2.1 ± 0.03	1.4 ± 0.04	20	++	0	0	0	0

No activity (−), halo or colony diameter < 5 mm (+), halo or colony diameter < 7 mm (++), halo or colony diameter < 10 mm (+++). Data are represented by means of at least three replicates ± SE at *p* ≤ 0.05 using LDS. PVK, Pikovskaya; IAA, indole-acetic acid; and CMC, carboxymethylcellulose.

## Data Availability

Not applicable.

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
