# Peer review of "Plant Growth-Promoting Bacterial Consortia as a Strategy to Alleviate Drought Stress in *Spinacia oleracea"

_microorganisms, 2022, doi:10.3390/microorganisms10091798_

Round 1

Reviewer 1 Report

The manuscript submitted to me for review entitled “Plant-Growth-Promoting Bacterial consortia as a strategy to alleviate drought stress in Spinacia oleracea” is a research paper dealing with the isolation and selection of plant growth-promoting bacteria, tolerant to drought, that can stimulate the spinach growth and development.

The title is well structured and corresponds to the manuscript’s content. The abstract systematizes the research idea and main findings.

Generally, the manuscript sections are well developed. It is well written and easy to understand.

In the first half, the introduction discusses all parts of the study. The second half of the Introduction is based on the history of the current research project even some conclusions appear. There aren’t defined clear study objectives for the present manuscript. I propose reorganising the section to clarify the hypothesis and objectives while previous findings and history could pass to the Methodology.

The section Materials and methods is developed in great detail that is positive from the point of view of research reproducibility. The analyses were performed using ANOVA, t-test or Duncan’s multiple comparison test with a significance level of p <0.05. Authors should carefully check the manuscript for abbreviations used where the term for the first time appears (for example, item 2.9).

Concerning results, they are well developed and in detail. The authors applied supplementary files also.

In my opinion, the discussion is well developed, comparing previous results with the authors’ one. The literature sources used in the manuscript are recent, as about 43% were published during the last five years.

The Conclusions are very general and could be improved.

Author Response

We would like to thank the reviewer for their suggestions that helped to improve our manuscript.

Point-by-point reply

The manuscript submitted to me for review entitled “Plant-Growth-Promoting Bacterial consortia as a strategy to alleviate drought stress in Spinacia oleracea” is a research paper dealing with the isolation and selection of plant growth-promoting bacteria, tolerant to drought, that can stimulate the spinach growth and development. 

The title is well structured and corresponds to the manuscript’s content. The abstract systematizes the research idea and main findings. 

Generally, the manuscript sections are well developed. It is well written and easy to understand. 

REPLY: Thanks. 

In the first half, the introduction discusses all parts of the study. The second half of the Introduction is based on the history of the current research project even some conclusions appear. There aren’t defined clear study objectives for the present manuscript. I propose reorganising the section to clarify the hypothesis and objectives while previous findings and history could pass to the Methodology.

REPLY: We have reorganized the introduction, adding some important information necessary to clarify our objectives.

The section Materials and methods is developed in great detail that is positive from the point of view of research reproducibility. The analyses were performed using ANOVA, t-test or Duncan’s multiple comparison test with a significance level of p <0.05.

REPLY: Thanks.

Authors should carefully check the manuscript for abbreviations used where the term for the first time appears (for example, item 2.9). 

REPLY: Done.

Concerning results, they are well developed and in detail. The authors applied supplementary files also. 

In my opinion, the discussion is well developed, comparing previous results with the authors’ one. The literature sources used in the manuscript are recent, as about 43% were published during the last five years. The Conclusions are very general and could be improved.

REPLY: Thanks. We have modified the Discussion and Conclusion section

Reviewer 2 Report

It is a document with very important and interesting results because it provides alternatives to mitigate the serious problem of water stress in plants. It is generally well written and presented. I think that by incorporating my suggestions the document can be published.

Author Response

We would like to thank the reviewer for their suggestions that helped to improve our manuscript.

Point-by-point reply

Plant-Growth-Promoting Bacterial consortia as a strategy to alleviate drought stress in Spinacia oleracea. It is a document with very important and interesting results because it provides alternatives to mitigate the serious problem of water stress in plants. It is generally well written and presented. I think that by incorporating my suggestions the document can be published.

REPLY: Thanks.

Abstract

Line 15: delete “to chemical fertilizers”
Line 19: delete “To this aim,”
Line 26: change fertilizer by “ameliorating water stress”

REPLY: Done.

Missing to include the spinach cultivar used

REPLY: The spinach cultivar was reported in the M&M section. As suggested by the Reviewer, it has been reported also in the abstract.

Introduction

First paragraph is too long, divide into two paragraphs

REPLY: Done.

The introduction lacks state of the art, it is important to include results of studies where these microorganisms have been used to mitigate water stress. I suggest eliminating the aspects related to fertilization, a lot of emphasis is placed on it, but the study is focused on mitigating water stress through many other mechanisms of microorganisms.

REPLY: We thank the reviewer for the useful suggestion. We have replaced the section on the aspects related to fertilization with new paragraphs on the state of the art of drought-tolerant PGPB activity for water stress mitigation(lines 53 to 77). We hope that the revised version properly addresses these issues.

Spinach is germinated in seedbeds and then the seedling is transplanted in the field, in the nursery there should be no carelessness with irrigation and water stress would be carelessness. Therefore, these aspects must be included in the introduction so that it is finally very clear that spinach is only used as a model plant for the germination phase.

REPLY: We have modified the text to clarify that Spinach has been used as a model plant (lines 81 to 84).

Line 80-83: I suggest that this paragraph be deleted and better included in discussion or conclusions

REPLY: The paragraph has been moved in the discussion and conclusion section (lines 546 to 549).

Materials and methods

What is the difference between the strain LMG9814 and the AGS108

REPLY: The two strains were isolated from different sources. Information on the strain source has been added in table 1.  

Line 133: include the spectroscope reference

REPLY: Done.

Line 140: include the spectrophotometer reference

REPLY: Done.

Line 183: include superscript of H2O2

REPLY: Done.

Line 203: include disinfection time

REPLY: Done.

Why 45 seeds? Are 45 seeds divided into three groups? Justify this number of seeds because ISTA

recommends 100 seeds per experimental unit

REPLY: In paragraph 2.4, it is reported that “the germination rate and efficiency were obtained from three independent experiments”, each performed on 45 seeds, for a total of 135 seeds. Nevertheless, this point was furtherly clarified as suggested by the Reviewer. We used this number of seeds similarly as previously reported in literature e.g. Avallone et al., 2022 and Makhaye et al., 2021.

Line 237: explain how length and weight were measured

REPLY: Pictures of the plates, together with a ruler as a scale, were taken from the same distance and the roots of the seedlings were measured by using ImageJ software, converting the length in pixels to centimeters. A new sentence has been added to explain the used method (lines 235 to 238)

Line 239: How many days after the start of the experiment were these measurements made?

REPLY: The measurements were performed after 27 days of growth. The information has been added to the text

Line 252-253: include superscripts

REPLY: Done.

Line 261: confusing, in this line it is indicated that mean and standard error are presented, but in line 256 mean and SD are indicated, is it depending on the experiment? Better to present only one, that is, either SD or SE.

REPLY: We agree with the Reviewer. We presented all data as mean and SD according to the comment.

Results

Line 269: name of bacteria in italics

REPLY: Done.

Line 306-307: better this phrase in the discussion

REPLY: The sentence has been moved, thank you for the suggestion.

Line 323-325: confused, *** is indicated in the figure, clarify in the figure

REPLY: thank you for the suggestion. Figure1 and its legend have been modified.

329: italics

REPLY: Done.

329-340: these sentences are about methodology, not about results, I suggest including what is not in the materials and methods chapter, and eliminating the rest.

REPLY: The sentences have been modified and moved to the relative paragraph of the M&M section.

380: name in italics

REPLY: Done.

Lines 402-414: include this information in materials and methods
point

REPLY: The information has been modified and included in the relative paragraph of the M&M section.

Figure 1. In addition to the anova, it is missing to include the results of the Duncan test, otherwise it is not possible to know the best strains. In C the statistics are missing, at least in the last sampling

REPLY: The Duncan test was not performed for this analysis as specified in the Material and Methods section. Statistics from Tukey’s test were added instead.

Figure 3: Same tip as Figure 1

REPLY: Done.

Figure 5. Improve the resolution of the figures. Indicate that it is under stress. Define C- and C2. Different letters in each average indicate differences according to Duncan's test (p<0.05)

REPLY: The resolution of the Figure5 has been improved and the legend modified to answer to raised points (f.i. the standard error has been replaced with the standard deviation and a new caption has been used)

Table S2: I suggest including statistics. SE was not included as indicated at the bottom of the table

REPLY: Done. A new version of table S2 has been submitted

Figure 1S: Asterisks indicate result of aNOVA? Or differences between control and stress? This is confusing, correct and indicate in the title of the figure the details of the symbols

REPLY: We agree with the Reviewer, the title of the figure has been changed as suggested.

Table 2. statistics are missing, in the title of the figure it says that LDS was used, but in the statistical analysis Duncan was indicated, correct

REPLY: The statistics were included as suggested. The Duncan test was applied for the physiological assays in Figure 5.

Discussion:

Why is Fv/fm in the control less than 0.8, was it also stressed?

REPLY: C- plants are control plants (not inoculated and not subjected to drought stress). All the Fv/Fm ratios reported in the graph are slightly lower than 0.8. However, it does not necessarily indicate a stress condition as such physiological values suggest a healthy status of the Photosystem II. Instead, our data highlight that the inoculation with the consortium improves the tolerance to drought stress.

Lines 445-457: Summarize this information because it seems state of the art, here is to use the information to explain your results. I suggest summarizing the results that are being included in this chapter, the idea is only to mention the most important results to connect with the discussion

REPLY: Both results and discussion sections have been revised following suggestions. Thank you

Better explain the results obtained, in some variables the explanations are very general, in the explanations try to use more all the measured variables.

REPLY: as written above, the discussion section has been modified.

The favorable result obtained in germination is not correctly discussed, for example, is it due to antioxidant activity? Increased hormone synthesis?

REPLY: We have modified the paragraph and speculated on the obtained data.

There are very long paragraphs.

REPLY: We have improved the text.